# Melody or Machine: Detecting Synthetic Music with Dual-Stream Contrastive Learning

**Arnesh Batra**[1]                                                arnesh23129@iiitd.ac.in
*Indraprastha Institute of Information Technology Delhi (IIIT-Delhi), India*

**Dev Sharma**[1]                                                dev23189@iiitd.ac.in
*Indraprastha Institute of Information Technology Delhi (IIIT-Delhi), India*

**Krish Thukral**[2]                                                krish.23fe10cse00679@muj.manipal.edu
*Manipal University Jaipur, Rajasthan, India*

**Ruhani Bhatia**[1]                                                ruhani23450@iiitd.ac.in
*Indraprastha Institute of Information Technology Delhi (IIIT-Delhi), India*

**Naman Batra**[3]                                                naman.batra.ug23@nsut.ac.in
*Netaji Subhas University of Technology (NSUT), Delhi, India*

**Aditya Gautam**[1]                                                aditya23043@iiitd.ac.in
*Indraprastha Institute of Information Technology Delhi (IIIT-Delhi), India*

**Reviewed on OpenReview:** *https://openreview.net/pdf?id=Ufwes0o2e3*

## Abstract

The rapid evolution of end-to-end AI music generation poses an escalating threat to artistic authenticity and copyright, demanding detection methods that can keep pace. While foundational, existing models like SpecTTTra falter when faced with the diverse and rapidly advancing ecosystem of new generators, exhibiting significant performance drops on out-of-distribution (OOD) content. This generalization failure highlights a critical gap: the need for more challenging benchmarks and more robust detection architectures. To address this, we first introduce Melody or Machine (MoM), a new large-scale benchmark of over 130,000 songs (6,665 hours). MoM is the most diverse dataset to date, built with a mix of open and closed-source models and a curated OOD test set designed specifically to foster the development of truly generalizable detectors. Alongside this benchmark, we introduce CLAM, a novel dual-stream detection architecture. We hypothesize that subtle, machine-induced inconsistencies between vocal and instrumental elements, often imperceptible in a mixed signal, offer a powerful tell-tale sign of synthesis. CLAM is designed to test this hypothesis by employing two distinct pre-trained audio encoders (MERT and Wave2Vec2) to create parallel representations of the audio. These representations are fused by a learnable cross-aggregation module that models their inter-dependencies. The model is trained with a dual-loss objective: a standard binary cross-entropy loss for classification, complemented by a contrastive triplet loss which trains the model to distinguish between coherent and artificially mismatched stream pairings, enhancing its sensitivity to synthetic artifacts without presuming a simple feature alignment. CLAM establishes a new state-of-the-art in synthetic music forensics. It achieves an F1 score of 0.925 on our challenging MoM benchmark, significantly outperforming the previous SOTA's 0.869 on the same dataset. This result demonstrates superior generalization to unseen generative models. Furthermore, CLAM scores 0.993 on the popular SONICS benchmark, confirming its effectiveness and setting a new performance standard.

# 1 Introduction

The rapid sophistication of end-to-end AI music generators has created a new class of synthetic media that is often indistinguishable from human created recordings. This development, encompassing everything from deepfake voice clones to fully synthesized songs, presents an urgent and evolving challenge. The core problem is not one of artistic judgment distinguishing, for example, avant-garde human compositions from AI but a technical one: reliably identifying the subtle, digital artifacts of the generative process itself. Failure to do so threatens to erode intellectual property rights, undermine artistic authenticity, and create a media environment where the line between human and machine creation is irreparably blurred.

Table 1: SpecTTTra OOD F1 Scores

| Dataset | Total Samples | F1 Score (%) |
|---|---|---|
| Riffusion | 7,057 | 53.46 |
| Yue | 5,278 | 68.80 |
| Voice clone | 1,166 | 50.94 |
| Suno 4 | 48 | 64.58 |

Table 2: Quantitative Comparison of Fake Song Datasets

| Dataset | Language | Avg. Length (sec) | # Algos | # Real Songs | # Fake Songs | Total Hours |
|---|---|---|---|---|---|---|
| **FSD** | Chinese | 216.00 | 5 | 200 | 450 | 26 |
| **SingFake** | Multi | 13.75 | − | 634 | 671 | 58 |
| **CtrSVDD** | Multi (no English) | 4.87 | 14 | 32,312 | 188,486 | 307 |
| **SONICS** | English | 176.03 | 5 | 48,090 | 49,074 | 4,751 |
| **MoM (ours)** | Multi (82% English) | 195.99 | 9 | 65,475 | 64,960 | 6,665 |

While foundational, early deepfake detection efforts were constrained by datasets with limited scale, diversity, and audio fidelity (Zang et al., 2024b;a; Liu et al., 2021; Tae et al., 2021; Xu et al., 2022). More recent benchmarks like SONICS (Rahman et al., 2025) represented an advancement, providing over 97,000 longer songs, including end-to-end synthetic tracks from platforms like Suno[1] and Udio[2] and introduced the SpecTTTra (Rahman et al., 2025) model for capturing long temporal dependencies. Nevertheless, SONICS exhibits drawbacks such as a male vocal bias, a lack of multilingual content, and irrelevant prompt attributes. The rapid progress in song generation models underscores the necessity for more diverse datasets and robust models capable of effectively handling out-of-distribution samples. . However, their utility is hampered by critical limitations: a reliance on only a few primary generation models and significant demographic biases (e.g., predominantly male, English language vocals). This lack of diversity creates a critical vulnerability. Models trained on these datasets learn to identify superficial, generator specific patterns, resulting in a dramatic failure of generalization when tested against out-of-distribution (OOD) content from unseen models. As we demonstrate, existing state-of-the-art detectors exhibit a sharp performance drop in these real world scenarios (see Table 1), proving them unreliable for practical application.

To confront this generalization crisis, we introduce MoM (Melody or Machine), a benchmark of 130,435 songs. Its key innovation is diversity, featuring a mix of open and closed-source generators (Yuan et al., 2025; Ziqian et al., 2025), a dedicated out-of-distribution (OOD) test set to ensure generalization, and 65,475 real songs including human covers to challenge models against natural stylistic variations. Unlike previous work, which uses the same models for training and testing, **MoM**'s training set includes **Suno v3.5**, **Suno v2**, **Diffrhythm** (Ziqian et al., 2025), and **Udio v1.5**, while the test set features **Riffusion (FUZZ-1.0)**[3], **Suno v4**, **Suno v3**, **Yue** (Yuan et al., 2025), and audio from **voice cloning methods**.

---

[1] https://www.suno.ai

[2] https://www.udio.com

[3] https://www.riffusion.com

We also propose CLAM (Contrastive Learning for Audio Matching), a novel dual-stream architecture that operates on mixed audio. Built on the hypothesis that AI disrupts a track's internal consistency, CLAM uses two pretrained encoders, MERT (Li et al., 2024) and Wave2Vec2 (Baevski et al., 2020), to generate parallel feature representations. A hybrid loss function then combines a binary classifier with a contrastive Triplet Loss (Schroff et al., 2015), which pulls representations of the same authentic track closer while pushing those from different tracks apart. This trains the model to recognize the inherent consistency of real recordings, a property we posit is disrupted by synthesis.

Our main contributions are as follows:

- **A New, Large-Scale Benchmark for Generalization (MoM):** We release the most diverse synthetic music dataset to date, featuring a wide array of generative models and a dedicated out-of-distribution test set designed to measure real-world robustness.

- **A Novel Dual-Stream Detection Model (CLAM):** We propose a new paradigm that analyzes parallel feature representations from mixed audio and is trained with a contrastive objective to identify subtle generative artifacts.

- **State-of-the-Art Performance and Analysis:** We demonstrate through extensive experiments that CLAM significantly outperforms existing models, achieving a new state-of-the-art F1-score of 0.925 on our challenging MoM benchmark, and provide a comprehensive analysis of the generalization failures of current detectors.

Table 3: Qualitative Comparison of Datasets

| Dataset | Fully Fake Songs | Text Lyrics Songs | Diverse Style Songs | Closed Source Models | Open Source Models | Multilingual Songs |
|---|---|---|---|---|---|---|
| **FSD (Xie et al., 2023)** | - | - | - | - | - | ✓ |
| **SingFake (Zang et al., 2024b)** | - | - | - | ✓ | ✓ | ✓ |
| **CtrSVDD (Zang et al., 2024a)** | - | - | - | ✓ | ✓ | ✓ |
| **SONICS (Rahman et al., 2025)** | ✓ | ✓ | - | ✓ | - | - |
| **MoM (ours)** | ✓ | ✓ | ✓ | ✓ | ✓ | ✓ |

## 2 Related Works

The rapid advancement of end-to-end generative models has led to a surge in AI-generated music, capable of producing entire songs with vocals, lyrics, instrumentation, and stylistic nuances. Detecting such synthetic compositions requires datasets and models that can capture both low-level audio artifacts and high-level musical coherence. In this section, we first survey the major public datasets designed for song deepfake detection, then turn to the spectrum of modeling strategies ranging from classical signal-processing approaches to spectogram-based architectures.

### 2.1 Audio Datasets for Synthetic Song Detection

Early efforts in synthetic song detection relied on small, narrowly scoped datasets, but recent benchmarks have begun to emphasize greater scale and diversity.

**SONICS** It is the current State-of-the-Art Dataset consisting of large collection of over 97,000 full-length tracks(4751 hours of audio) evenly split between real recordings and synthetic songs generated by platforms

such as Suno (v2–v3.5) and Udio (v32, v130). Crucially, SONICS's songs are quite long (average of 176 seconds), supporting the modeling of long-range musical and lyrical patterns. It also includes the text lyrics of the songs, which can aid future research. However, SONICS also has clear limitations since it consists only of English songs (mostly featuring male vocals) with audio processed at a fixed 16 kHz rate, and its synthetic tracks come exclusively from two AI models (Suno and Udio), with the Half Fake tracks being only generated from one model- which means one compositional scenario (real lyrics with Udio) is missing, narrowing the variety within that category.

**Other Datasets**  SingFake offers 58 hours of paired real and synthetic vocal clips across five languages and 40 singers, using real instrumental backings to aid in artifact detection. However, it lacks fully synthetic lyrics or instrumentals. CtrSVDD builds upon this by providing 308 hours of controlled SVDD content (over 220,000 clips), enabling fine-grained manipulation through parameterized synthesis and metadata through it, centering on vocals rather than full-song generation. This focus on partial clips or vocals, while useful, may not capture the long-range compositional narrative and complex interactions between elements present in full-song tracks, which is a key challenge for detection. Meanwhile, FSD (Fake Song Detection) (Xie et al., 2023) is a Chinese language benchmark comprising 200 real and 450 fake songs produced via five modern synthesis and conversion techniques. Its difficulty for speech-based detectors underscores the need for music-specific modeling, though its limited linguistic and methodological scope hinders broader applicability.

Although prior datasets have laid important groundwork for synthetic music detection, they often lack the diversity in generation pipelines and audio characteristics needed for models to generalize effectively to emerging AI content. Our contribution includes the Melody or Machine (MoM) dataset, specifically designed with extensive source and variation coverage to bridge this gap.

## 2.2 Modeling Approaches

Detecting AI-generated music spans a broad range of techniques, from classical signal transforms to deep neural architectures. **Early methods** (Pour et al., 2014; Wani et al., 2024a;b; Pham et al., 2024; Yi et al., 2023) convert raw waveforms into time–frequency representations such as STFT, CQT, MFCCs, or gammatonegrams. These are typically fed into convolutional or recurrent networks to identify synthesis artifacts. While effective at highlighting low-level inconsistencies, these approaches often fail to capture long-term musical structure and are sensitive to genre and production variations.

To address these limitations, **recent methods incorporate long-context modeling** using attention mechanisms and spectro-temporal tokenization. For example, models like SpecTTTra (Rahman et al., 2025) (SOTA on SONICS) decompose spectrograms into patch-based tokens and apply transformers or memory-augmented RNNs to model global coherence in lyrics, rhythm, and instrumentation. These techniques show promise in detecting higher-order anomalies but risk overfitting to specific generator signatures, limiting their robustness across domains.

Another line of work leverages **self-supervised audio encoders** (Sharma & Gupta, 2025; Guo et al., 2024; Phukan et al., 2024a;b) such as MERT, wav2vec2, HuBERT (Hsu et al., 2021), and Audio Spectrogram Transformer (AST) (Gong et al., 2021). These models are pretrained on large speech or music corpora to learn rich acoustic representations. When fine-tuned for deepfake detection, they often outperform purely supervised models. However, the gap between speech and music, characterized by wider pitch ranges, melodic structures, and vocal expression, necessitates domain-specific adaptation or joint pretraining across modalities.

In summary, the field has progressed from handcrafted audio features to sophisticated neural architectures with self-supervised learning and structured modeling. However, detecting fully end-to-end generated songs blending vocals, lyrics, instrumentation, and style remains an open challenge. Our work builds on these insights by leveraging pretrained encoders combined with contrastive training, achieving state-of-the-art performance in AI-generated music detection.

## 3 Methodology

### 3.1 The Melody or Machine (MoM) Dataset

This section details our methodology for creating a benchmark to detect specific types of AI-generated music, ranging from partial manipulations like voice cloning to fully synthetic, end-to-end generated songs. To address the critical generalization failures of existing detectors, we developed the Melody or Machine (MoM) dataset, a large-scale benchmark designed to reflect this diversity. MoM provides 130,435 audio tracks organized into three operational tiers: *Real*, *Fully Fake*, and *Mostly Fake*, which together enable a nuanced evaluation of detection models. The dataset incorporates a wide array of both proprietary and open-source generators, including Suno (v2–v3.5), Udio (v1.5), Riffusion, Diffrhythm (Ziqian et al., 2025), and Yue (Yuan et al., 2025).

#### 3.1.1 Real Songs

The *Real* subset comprises 65,475 human-created tracks, providing a robust ground truth for authentic audio. This includes 47,971 original songs sourced from YouTube, with metadata linked from the Genius Lyrics Dataset (Lim & Benson, 2021), and 17,504 high-quality human-performed covers. The inclusion of covers is a key design choice, as it challenges models to distinguish genuine AI artifacts from the natural variations in timbre, arrangement, and style inherent in human creativity.

#### 3.1.2 Fully Fake Songs

This tier contains 53,922 tracks generated via a structured prompting pipeline designed to ensure musical relevance and stylistic diversity, moving beyond simple random generation. To achieve this, we employed three distinct prompting strategies.

- Type A: Inspired by Existing Songs - To simulate a common human creative process, prompts were generated by reimagining existing song titles in a new musical style, with genres sampled from a curated list of 163 options derived from the FMA dataset (Defferrard et al., 2018; 2017).

- Type B: Curated by Musical Attributes - We created a structured taxonomy of musical features: *genre, mood, tempo, key, instrumentation*, and *production style*. We then programmatically generated thousands of unique attribute combinations. These were refined using Gemini 2.0 Flash (Google, 2025) to translate them into fluent, natural-language prompts suitable for generation systems.

- Type C: Community-Sourced - To capture "in-the-wild" usage, we sourced high-frequency prompts directly from public, state-of-the-art generative music platforms, enhancing the dataset's robustness against practical edge cases and organic user trends.

#### 3.1.3 Mostly Fake

This subset simulates partial synthetic manipulation, a common real-world scenario.

- Type A: Real Lyrics + AI Audio - We sourced timestamped lyrics from commercial songs and used a fine-tuned BERT-based classifier (Devlin et al., 2019) to predict a stylistically appropriate genre for the lyrical content. The lyrics and predicted genre were then fed to models like Diffrythm and Yue to generate coherent audio.

- Type B: Voice Cloning - This subset of approximately 1,000 samples addresses another prevalent deepfake scenario: AI-generated vocal covers. Here, a known artist's vocal timbre is synthetically applied to a new performance (using different lyrics or instrumentals). These samples, sourced from public platforms, represent contemporary voice cloning practices and test a model's ability to detect vocal synthesis specifically.

Table 4: This table shows the number of audio samples used for training/validation and testing, organized by model. Models marked in **red** are **closed-source** (e.g., Suno, Udio, Riffusion, Voice Clone) and models marked in **blue** are **open-source** (e.g., Yue and Diffrythm).

| Model | Train / Validation | Test |
|---|---|---|
| Suno 3.5 | 23695 | – |
| Udio 1.5 | 19500 | – |
| Diffrythm | 4606 | – |
| Suno 2 | 110 | – |
| Suno 1 | – | 48 |
| Suno 3 | – | 3512 |
| Riffusion | – | 7057 |
| Yue | – | 5278 |
| Voice Clones | – | 1166 |
| **Total** | **47911** | **17061** |

### 3.1.4  Dataset Evaluation

The MoM dataset improves on prior benchmarks like SONICS with broader genre, language, and vocal diversity, enabled by a more expressive prompt pipeline and a mix of open and closed-source models. It supports realistic, end-to-end song generation with coherent vocals and instrumentation, reflecting new AI music trends.

To quantitatively measure and benchmark the perceptual quality of the generated audio, we established a human evaluation platform. The evaluation was conducted with a **general, non-expert participant demographic, primarily university students** and their networks, to gauge broad perceptual quality relevant to real-world content moderation scenarios. Participants performed blind, head-to-head comparisons where the **A/B pairings were randomized** across the entire pool of models to produce a general perceptual leaderboard.

The central question posed to participants was "**Which song sounds better overall?**". To guide this subjective assessment and ensure a composite perceptual judgment, we instructed evaluators to base their choice on four key aspects of the audio:

- **Overall Audio Quality:** This included listening for clarity, an absence of distortion or strange artifacts, and the overall fidelity of the track.

- **Realism:** Judges were asked to consider how natural and convincing the instruments, vocals, and the complete soundscape felt.

- **Musicality:** An optional factor, this pertained to whether the melody was catchy, the harmony was pleasing, and the musical structure was coherent.

- **Lyrics:** Also optional and depending on the sample, this involved evaluating if the lyrics were clear, coherent, and well-integrated into the song.

These human judgments were then aggregated into a quantitative ranking using the ELO rating system, adapted from chess, which dynamically scores each model based on its win/loss outcomes. The ELO rankings were calculated **exclusively based on the mandatory "Which song sounds better overall?" question**. The resulting ELO score thus reflects the overall perceptual preference defined by our multifaceted voting instructions (clarity, realism, fidelity, and optionally musicality/lyrics).

Our evaluation shows that MoM's synthetic tracks achieve a higher average ELO score (1016.35) than those in the SONICS benchmark (894.60), indicating their superior perceived quality. Most notably, the perceived quality of our top synthetic models, like Riffusion (ELO 1105.58) and Udio (ELO 1093.34), was so high that they were ranked above authentic human-created songs (ELO 1032.84) by listeners. This underscores

that the line between human and machine creation is blurring, motivating the need for more sophisticated detection methods. Our dataset emphasizes generalization by incorporating a curated test set composed of diverse out-of-distribution (OOD) samples, including Riffusion (FUZZ-1.0), Yue, Suno 3, Suno 4, and Voice Clones. In contrast, prior works primarily evaluate on variants of similar models such as Suno and Udio, which leads to overly optimistic results that do not reflect real-world generalization as shown in Table 1.

Table 5: Human evaluation results using ELO-based ranking. Participants voted on which sample 'sounds better,' considering clarity, absence of artifacts, overall fidelity, realism, and optionally musicality and lyrical coherence. The resulting ELO scores thus capture a composite perceptual preference. See Appendix for details.

| Model | ELO Score |
|---|---|
| Riffusion (ours) | 1105.58 |
| Udio (ours) | 1093.34 |
| Real | 1032.84 |
| Suno (ours) | 1013.76 |
| Voice Clones (ours) | 1007.23 |
| Yue (ours) | 958.37 |
| Diffrythm (ours) | 934.14 |
| Suno (SONICS) | 901.75 |
| Udio (SONICS) | 887.44 |
| **Average (Ours)** | **1016.35** |
| **Average (SONICS)** | **894.60** |

## 3.2 CLAM: A Probabilistic Approach to Learning Audio Authenticity

We introduce CLAM (Contrastive Learning for Audio Matching), an architecture designed not to find simple spectral artifacts, but to learn the holistic signature of musical authenticity. CLAM operates on the full, mixed audio signal, a key feature for practical, real-world applications.

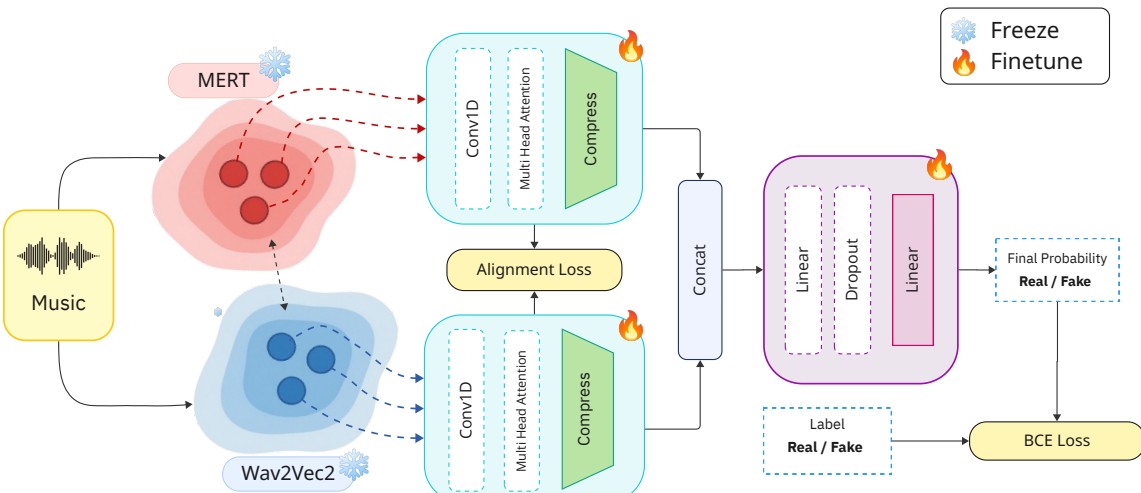

Figure 1: Overview of the CLAM (Contrastive Learning for Audio Matching) architecture. This two-encoder model processes instrumental and vocal embeddings (from sources like MERT and Wav2Vec2), incorporates a Weighted Cross-Aggregation (WCA) module, and a dual-loss objective.

### 3.2.1 Core Intuition: A Probabilistic View on Musical Authenticity

An authentic musical recording is a sample from a complex, structured joint probability distribution, $P(V, M)$, where $V$ and $M$ represent the vocal and instrumental elements. This distribution is governed by the physics of acoustics and the grammar of music theory. Our central hypothesis is that AI generators learn a flawed approximation, $\hat{P}(V, M)$. Our goal is to train a detector that is maximally sensitive to the Kullback-Leibler (KL) divergence (Hershey & Olsen, 2007) between these two distributions:

$$D_{KL}(P||\hat{P}) = \sum_{x \in (V,M)} P(x) \log \frac{P(x)}{\hat{P}(x)} \tag{1}$$

We posit this divergence is most pronounced in the subtle, high-order statistical dependencies between musical elements.

**Manifestations of the Modality Gap.** This discrepancy, or modality gap, manifests in musically significant ways. For instance, AI generators may break the assumption of conditional dependence; a human singer's vocal timbre changes with pitch, but an AI might model these as conditionally independent given the melody, creating an unnatural sound. Similarly, the rhythmic pocket of a human performance, a structured deviation from a perfect grid, is a complex dependency that AI often fails to replicate, resulting in rhythms that are either sterilely quantized or exhibit unstructured, random timing.

**The Authenticity Manifold.** Geometrically, we frame this using the *Manifold Hypothesis*. We posit that feature vector pairs $(\mathbf{d_m}, \mathbf{d_v})$ extracted from authentic songs lie on or near a low-dimensional *authenticity manifold*, $\mathcal{A}$, embedded within a high-dimensional joint feature space $\mathbb{R}^d \times \mathbb{R}^d$. Synthetic songs, drawn from the flawed distribution $\hat{P}$, produce feature pairs that lie off this manifold. CLAM's novelty is its design to first learn the geometry of $\mathcal{A}$ and then identify any signal that deviates from it.

### 3.2.2 Dual-Stream Feature Projections

To perceive this manifold, CLAM projects the mixed audio signal, $S$, onto two distinct, complementary feature subspaces using pretrained encoders acting as projection functions, $\phi_M$ and $\phi_V$:

- *Music-Aware Projections ($\phi_M$):* We use MERT (Li et al., 2024) to generate a music-centric view, $\mathbf{d}_{\text{music}} = \phi_M(S)$, capturing structural information like harmony and rhythm.

- *Timbral-Acoustic Projections ($\phi_V$):* We use Wave2Vec2 (Baevski et al., 2020) to generate a view emphasizing fine-grained timbral and articulatory textures, $\mathbf{d}_{\text{vocal}} = \phi_V(S)$.

### 3.2.3 Weighted Cross-Aggregation (WCA) Module

The WCA module distills the raw, multi-layered encoder outputs into compact, fixed-size embeddings.

**1. Learned Layer Aggregation.** To create an optimal representation, we compute a learnable weighted sum across all $L$ encoder layers. For a set of layer outputs $\{\mathbf{M}^{(i)}\}_{i=1}^{L_M}$, the aggregated map is $\overline{\mathbf{M}} = \sum_{i=1}^{L_M} \mathbf{w}_i \mathbf{M}^{(i)}$, where the weights $\mathbf{w}_i$ are the learned parameters of a 1D convolution (O'Shea & Nash, 2015) with a kernel size of 1.

**2. Intra-Stream Self-Attention.** Before fusion, a multi-head self-attention mechanism (Vaswani et al., 2023) refines each aggregated stream. By computing attention scores between Query, Key, and Value projections of the input sequence, this step builds a context-aware representation that focuses on the most salient musical or vocal phrases.

### 3.2.4 Learning the Manifold's Geometry via Contrastive Metric Learning

Our learning objective is a hybrid of discriminative classification and probabilistic metric learning. A standard Binary Cross-Entropy (BCE) loss on the concatenated embeddings drives the final classification. We

supplement this with a powerful inductive bias via the *Triplet Loss* (Schroff et al., 2015), which acts as a geometric regularizer. This approach can be formally understood through the lens of an energy-based model (EBM). We define an energy function $E_\theta(\mathbf{d_m}, \mathbf{d_v})$ parameterized by the model weights $\theta$, which should assign low energy to authentic, internally consistent pairs and high energy to all other pairs. The goal is to learn $\theta$ such that the Gibbs-Boltzmann distribution (Rowlinson*, 2005), assigns high probability to authentic pairs. The Triplet Loss is a powerful method to shape this energy surface without computing the intractable partition function $Z_\theta$. It enforces the margin-based constraint that the energy of an authentic, matched pair (positive) must be lower than the energy of a mismatched pair (negative) by at least a margin $\alpha$:

$$E_\theta(\mathbf{a}, \mathbf{p}) + \alpha < E_\theta(\mathbf{a}, \mathbf{n}) \tag{2}$$

where the energy $E$ is instantiated as the squared Euclidean distance. The loss function is thus:

$$\mathcal{L}_{\text{triplet}} = \max(0, E_\theta(\mathbf{a}, \mathbf{p}) - E_\theta(\mathbf{a}, \mathbf{n}) + \alpha) \tag{3}$$

We apply this loss *only to real songs* in a batch, where for a given real song $i$:

- *Anchor* ($\mathbf{a}$): The music-centric embedding, $\mathbf{d}_{\text{music}}^{\text{real}}[i]$.

- *Positive* ($\mathbf{p}$): The vocal-centric embedding of the *same* song, $\mathbf{d}_{\text{vocal}}^{\text{real}}[i]$.

- *Negative* ($\mathbf{n}$): The vocal-centric embedding from a *different* real song $j$.

This objective forces the two views of any real song to have low energy, making the authenticity manifold $\mathcal{A}$ a compact, low-energy region. The total loss,

$$\mathcal{L}_{\text{total}} = \mathcal{L}_{\text{BCE}} + \lambda \cdot \mathcal{L}_{\text{triplet}} \tag{4}$$

jointly optimizes for classification accuracy and a latent space structure that reflects the probability distribution of authentic data. The contrastive loss acts as a strong regularizer, simplifying the classifier's task to separating the low-energy manifold from the high-energy space around it.

## 4 Experiments and Results

### 4.1 Experimental Setup

All models were trained on an NVIDIA RTX 4060 Ti 16GB GPU. We used the AdamW optimizer (Loshchilov & Hutter, 2019) with a learning rate of 1e-4, a batch size of 128, and an embedding dimension of 512 for 50 epochs. For reproducibility, all experiments were run with 5 seeds and the results provided are average across those them, and all code is provided in the supplementary material. Audio samples were trimmed to 90 seconds and resampled to 24 kHz to balance computational load with the preservation of high-frequency details.

Our primary evaluation metric is the **F1 score**, chosen for its robustness in measuring performance on imbalanced or challenging OOD test sets where simple accuracy can be misleading (Christen et al., 2023). While multiple runs for statistical significance tests were computationally prohibitive given the dataset's scale, we provide detailed ablation studies to rigorously validate our contributions. For experiments involving the contrastive loss, the weight $\lambda$ was held at 0.5 based on preliminary validation performance which was carried out in the range of 0.1 to 1 values of $\lambda$.

### 4.1.1 Main Results on the MoM Benchmark

The primary goal of our work is to address the failure of current models on diverse, out-of-distribution data. We therefore present our main results on the challenging MoM benchmark, which was specifically designed to test this generalization capability. As shown in Table 7, the previous state-of-the-art model, SpecTTTra, achieves an F1 score of 0.869 on MoM. Other recent baselines, MiO (Phukan et al., 2024a) and Poin-HierNet (Yang et al., 2025), achieve 0.872 and 0.896, respectively, highlighting the difficulty of the benchmark.

Table 6: F1 Score performance on the SONICS dataset.

| Method | F1 Score |
|---|---|
| SpecTTTra-$\alpha$ (120 sec) | 0.972 |
| CLAM (No Alignment Loss) | 0.987 |
| CLAM (Triplet Loss) | **0.993** |

Table 7: Performance comparison on MoM dataset.

| Method | Accuracy (%) | F1 Score (%) |
|---|---|---|
| *State of the Art Models* | | |
| SONICS SpecTTTra-$\alpha$ (Rahman et al., 2025) | 87.2 | 86.9 |
| MiO (Phukan et al., 2024a) | 88.4 | 87.2 |
| Poin-HierNet (Yang et al., 2025) | 90.3 | 89.6 |
| *Unimodal Only* | | |
| AST (No WCA) | 81.6 | 80.1 |
| HUBERT (No WCA) | 82.0 | 78.9 |
| MERT (No WCA) | 83.5 | 81.3 |
| Wav2Vec2 (No WCA) | 82.7 | 80.2 |
| MERT Only | 86.1 | 85.3 |
| Wav2Vec2 Only | 83.1 | 84.6 |
| *Multimodal (No Alignment Loss)* | | |
| CLAM (No Alignment Loss) | 88.8 | 87.5 |
| *Multimodal + Alignment Losses* | | |
| CLAM (MSE Loss) | 90.1 | 89.9 |
| CLAM (Huber Loss, $\delta$=1.0) | 90.3 | 90.1 |
| CLAM (Cosine Loss) | 90.8 | 90.3 |
| **CLAM (Triplet Loss)** | **93.1** | **92.5** |

In contrast, our proposed **CLAM model with Triplet Loss achieves a new state-of-the-art F1 score of 0.925**. This significant improvement of nearly 6 points over SpecTTTra and 3 points over the strong Poin-HierNet baseline demonstrates the superior generalization capabilities of our dual-stream, contrastive approach when faced with a true OOD challenge.

### 4.1.2 Ablation Studies

To deconstruct the sources of this performance gain and validate our design choices, we conducted a series of ablation studies.

The unimodal baselines, using only MERT or only Wave2Vec2, achieve F1 scores of 0.853 and 0.846, respectively (Table 7, MERT Only/Wav2Vec2 Only). Our dual-stream CLAM model, even without the alignment loss, achieves an F1 score of 0.875. This substantial improvement confirms that analyzing the audio through two complementary feature streams captures richer information than either stream can alone, validating our core architectural hypothesis.

Another critical question is whether the alignment loss component provides a meaningful benefit. By comparing the CLAM model without this loss (0.875 F1) to the full model with Triplet Loss (0.925 F1), we see a clear and significant performance increase. This demonstrates that explicitly training the model to structure the embedding space by pulling representations of the same real song together and pushing others apart is highly effective. It confirms our intuition that modeling the internal consistency of authentic recordings is a powerful signal for detecting AI-generated content.

We validated our performance gains using two-sided McNemar's tests, with full methodological details and contingency tables provided in the Appendix. The results show that our best model, CLAM (Triplet Loss), achieves highly significant improvements over both SpecTTTra and Poin-HierNet ($p < 0.001$). The dual-stream CLAM (No Alignment Loss) also significantly outperforms the strongest unimodal baseline, MERT Only ($p < 0.01$). Moreover, incorporating the alignment loss yields a highly significant boost, with CLAM (Triplet Loss) outperforming CLAM (No Alignment Loss) ($p < 0.001$). Overall, these tests confirm that our architectural choices and the Triplet alignment loss lead to robust, statistically reliable gains. Detailed results are included in the Appendix.

### 4.2 Performance on the SONICS Benchmark: A Saturation Test

To ensure our model also performs well on existing benchmarks, we evaluated it on the SONICS dataset. As shown in Table 6, CLAM achieves a near-perfect F1 score of 0.993, marginally outperforming the non-contrastive variant (0.987) and the SpecTTTra-$\alpha$ model (0.972).

However, the near-perfect scores achieved by multiple top models on this dataset indicate that it has become "saturated." It is no longer sufficiently challenging to differentiate the capabilities of advanced architectures. This observation further underscores the novelty and necessity of our more diverse and challenging MoM benchmark for driving future progress in the field.

## 5 Discussion and Conclusion

In this paper, we confronted the critical challenge of out-of-distribution generalization in AI-generated music detection. We introduced **MoM**, the most diverse public benchmark to date, specifically designed with a dedicated OOD test set to drive the development of truly robust models. MoM's inclusion of varied generators, multiple languages, and human cover songs provides a more realistic and challenging evaluation standard for the field. Alongside this benchmark, we proposed **CLAM**, a novel dual-stream architecture. Instead of relying on fragile, generator-specific artifacts, CLAM is designed to detect subtle inconsistencies between parallel feature representations of a single mixed audio track. Our experiments provide strong support for this approach: the dual-stream model consistently and significantly outperforms unimodal baselines. Furthermore, the use of a contrastive Triplet Loss measurably improves performance by encouraging the model to learn the internal consistency of authentic recordings. This method provides a more principled way to identify synthetic content than relying on simple spectral pattern matching. By achieving a new state-of-the-art F1 score of 0.925 on the challenging MoM benchmark, this work not only delivers a superior detection model but also establishes a new, more rigorous methodology for evaluating synthetic music detectors. By releasing MoM, we hope to catalyze a shift in focus from in-domain performance to the far more critical goal of real-world generalization. Our Main contributions is filling the gap in the existing works by making a novel dataset and showing that understanding the semantics of music are very important in this problem.

## 6 Limitations

Our primary limitation stems from the relentless pace of innovation in AI music generation. The generative models used in this study represent the state-of-the-art today, but new architectures are constantly emerging. This "arms race" means any detection model, including ours, faces a continuous risk of obsolescence as new, more sophisticated generators produce music with different, more subtle artifacts. Consequently, the MoM dataset will require periodic updates and our model will need retraining to remain effective against future threats. A second significant challenge is the computational cost associated with our model's size and complexity. Additionally, our dataset is predominantly English (82%), and the model's behavior on other forms of computer-generated music (e.g., algorithmic composition) is untested.

## 7 Broader Impacts

The MoM dataset and the CLAM model create noteworthy positive societal outcomes through a greater capacity for the detection of AI-created music. For example, content platforms, rights holders, as well as forensic analysts, gain the ability to identify synthetic tracks. This supports the security of artist intellectual property and the maintenance of trust in music distribution. MoM's detailed annotations plus song representations with attention to multilingualism and gender balance give support to the creation of moderation tools in the future. These tools can offer more fairness and inclusivity across the music industry. At the same time, a large, good corpus of actual plus synthetic songs could help in the training of deepfake generators that appear more real.

The focal point of the project is to label music on popular platforms, so that the music is presented to the listeners with full transparency about the production of the music they are consuming, making them aware of authenticity as well as the ability of AI. Music artists may benefit from this labeling as well, as it prevents faking AI generated music to be posed as human made which provides an unfair advantage by displaying a level of human element to the listener, this is similar to the issues faced by graphic designers who had their livelihood affected by the advancements in AI image generators which would have been prevented to an extent, had a labelling system been introduced sooner.

Full dependency on automated detectors presents a risk of false positives or unfair treatment of genres or vocal styles not well represented in the data as well as potentially harmful to smaller artists if their music is incorrectly labelled as synthetic. Another point of potential issues affects artists who either unknowingly use samples generated with AI or those who are physically incapable of some part of production- either vocals or instrumentals and want to use AI to complete their work. To lower those risks, a release strategy for the full dataset with controls is a good idea.

## 8 Ethics Statement

Our research was conducted with careful consideration for ethical implications, focusing on responsible data sourcing and transparency.

**Data Sourcing and Licensing** The *Real* songs and *Mostly Fake Type B* (voice cloning) samples are sourced from YouTube. To respect copyright, we do not host or distribute this audio; we provide only the original YouTube links for research replication. All AI-generated songs created for the MoM dataset will be released on Hugging Face under a Creative Commons CC BY-NC 4.0 license, permitting non-commercial research use. We acknowledge that some data was generated using publicly accessible commercial AI tools. This work uses the generated output for non-commercial, academic research with the explicit goal of advancing deepfake detection and promoting authenticity a protective and beneficial application. This aligns with established research practices for creating evaluation benchmarks (Shin et al., 2024; Yang et al., 2023; Wang et al., 2025). We recognize the evolving legal landscape surrounding AI-generated content and have acted in good faith to create a resource for positive research outcomes.

**Dual-Use Mitigation** There is an inherent dual-use risk that a dataset of synthetic music could be used to train more effective generative models. To mitigate this, the full MoM dataset will be released under a controlled access policy to verified researchers, prioritizing its use for defensive applications like deepfake detection.

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

# A    Technical Appendices and Supplementary Material

## A.1    Dataset Description

For our experiments, we curated a diverse dataset comprising both real and synthetic songs sourced from several state-of-the-art generative audio models. The dataset spans a wide variety of generation techniques, quality levels, and stylistic characteristics, enabling comprehensive evaluation of detection and generalization performance.

### A.1.1    Composition and Distribution

The synthetic portion of the dataset includes samples from the following models:

- **Suno v2** – 110 samples

- **Suno v3** – 3,512 samples *(Test only)*

- **Suno v3.5** – 23,695 samples

- **Suno v4** – 48 samples *(Test only)*

- **Udio v1.5** – 19,500 samples

- **Diffrythm** – 4,606 samples

- **Riffusion** – 7,057 samples *(Test only)*

- **Yue** – 5,278 samples *(Test only)*

- **Voice Cloning** – 1,166 samples *(Test only)*

**Fake-Type Categorization.** We further categorize the synthetic songs based on the degree of AI synthesis:

- **Fully Fake:** Suno (v2, v3, v3.5, v4), Udio v1.5, Riffusion

- **Mostly Fake:** Yue, Diffrythm, Voice Cloning

This structured labeling allows systematic evaluation under varying levels of generative realism.

## A.2 Human-AI Perceptual Benchmark

To benchmark our detection models against human perceptual judgment, we developed a blind listening evaluation platform.

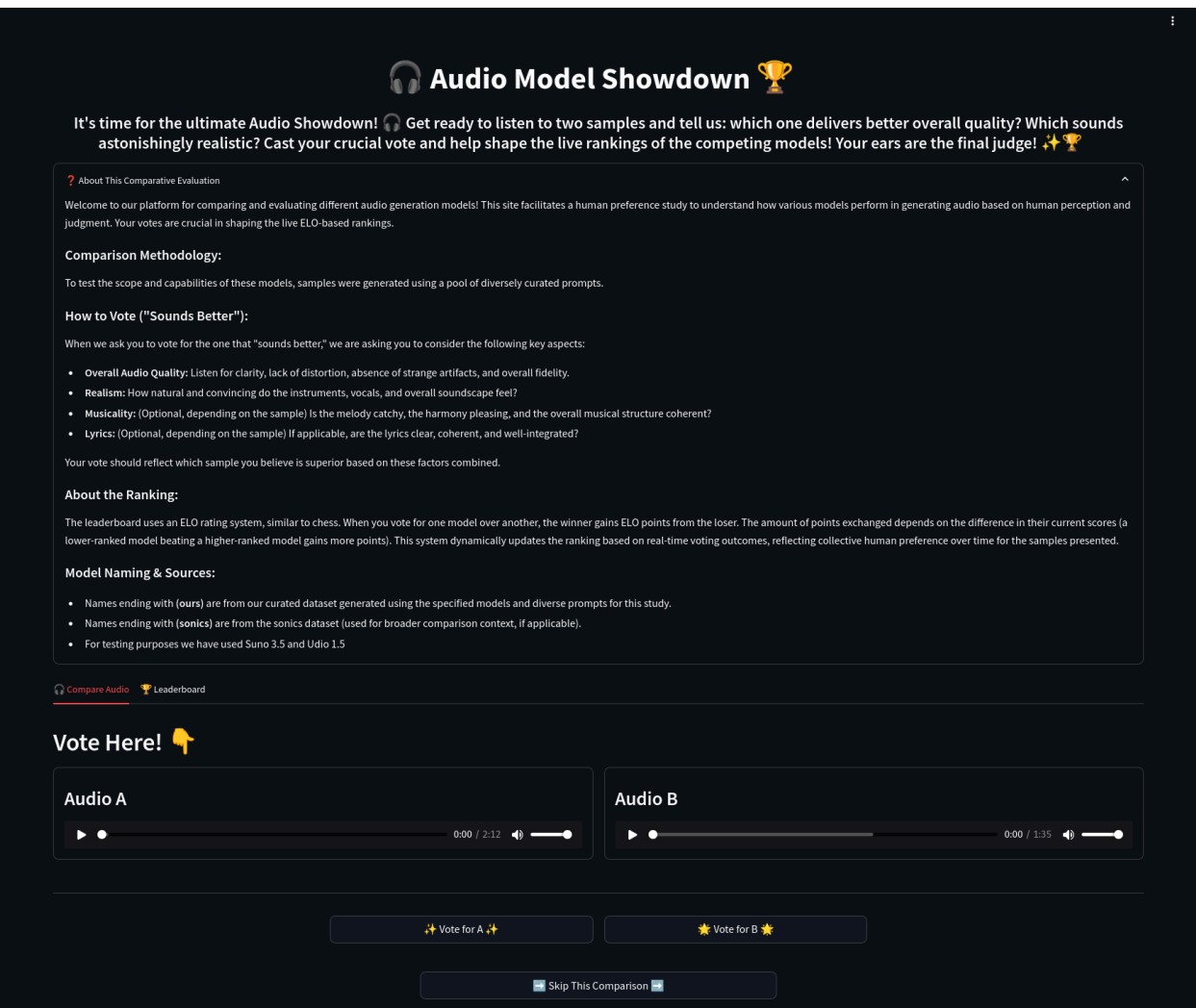

Figure 2: Interface of the Song Arena platform on Huggingface.

### A.2.1 Evaluation Setup

Participants are presented with randomized audio pairs and asked: *"Which song sounds more realistic?"* Each pair falls into one of two categories:

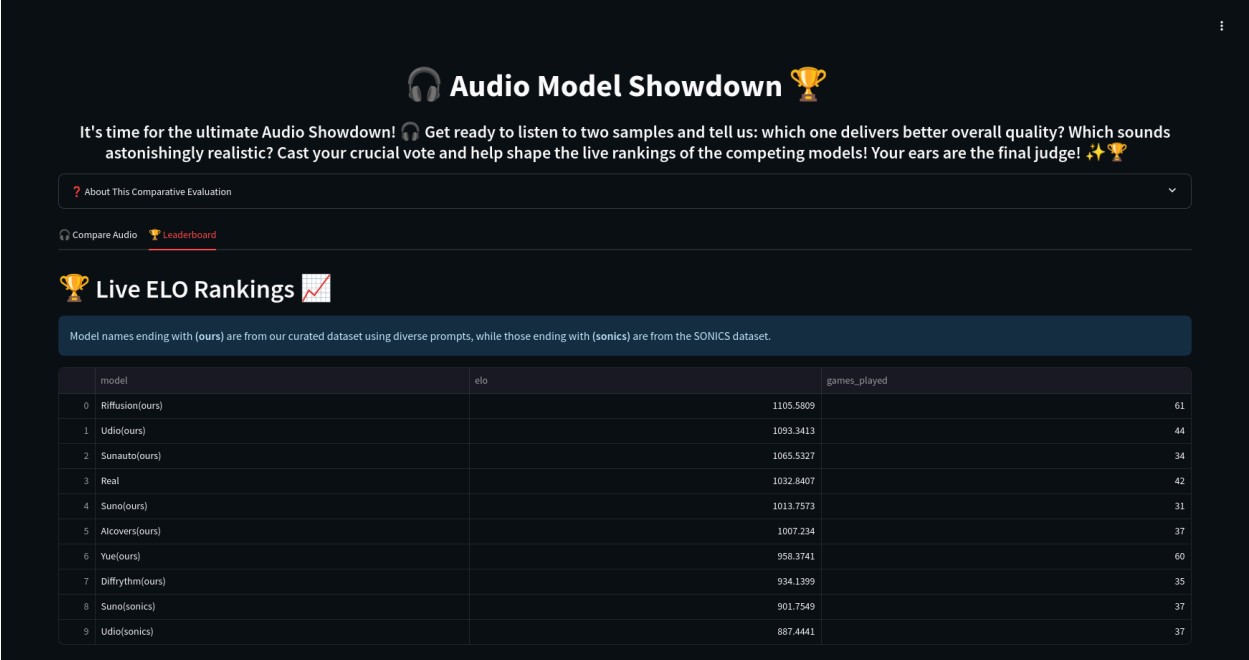

Figure 3: Leaderboard based on Elo scores from human evaluations on the Song Arena platform. Higher scores indicate stronger preference by listeners.

- **Intra-Dataset:** Both songs are drawn from the same dataset but different generation models.

- **Inter-Dataset:** Songs are drawn from entirely different datasets (e.g., MoM vs. external benchmarks like SONICS).

### A.2.2 Model Elo Ranking from our website

To quantitatively assess perceived audio realism from human evaluations, we compute an Elo rating based on aggregated pairwise comparisons our platform. Higher scores indicate greater preference by listeners. Our proposed Riffusion-based model achieves the top human realism score.

Table 8: Elo Scores from Human Evaluation

| Model | Elo Score |
|---|---|
| Riffusion (ours) | 1105.58 |
| Udio (ours) | 1093.34 |
| Real | 1032.84 |
| Suno (ours) | 1013.76 |
| Voice Clones (ours) | 1007.23 |
| Yue (ours) | 958.37 |
| Diffrythm (ours) | 934.14 |
| Suno (SONICS benchmark) | 901.75 |
| Udio (SONICS benchmark) | 887.44 |

**Interpretation:** The Elo-based analysis reveals that multiple synthetic models (notably our versions of Riffusion and Udio) surpassed human-composed songs in perceived realism. This demonstrates the increasing difficulty of AI song detection and motivates the need for sophisticated modality-alignment-based detection methods.

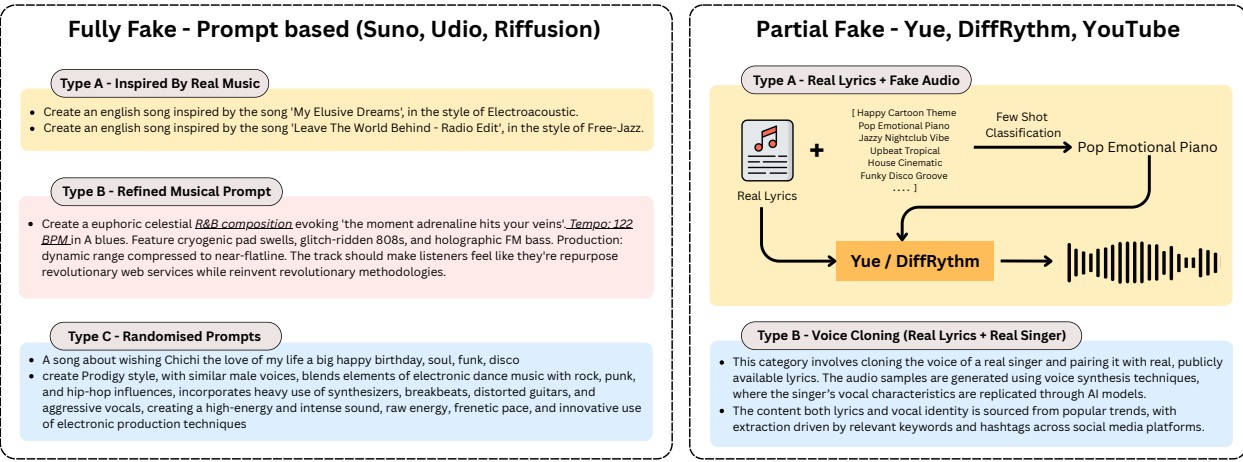

Figure 4: Distribution of AI-generated songs in our dataset.

## A.3 Prompt Metadata and Style Attributes

Each generated song is associated with prompt metadata and stylistic attributes which provide a rich representation of musical intent. These structured tags help in conditioning models, analyzing generalization, and understanding detection robustness.

### A.3.1 Genre List

*pop, rock, hip-hop, R&B, EDM, jazz, blues, country, metal, punk, folk, classical, ambient, lo-fi, trap, drum and bass, dubstep, synthwave, vaporwave, house, techno, trance, reggae, dancehall, afrobeats, k-pop, j-pop, gospel, funk, soul, indie rock, indie pop, math rock, psychedelic, experimental, industrial, noise, electroacoustic, bossa nova, samba, latin pop, grime, phonk, drill, hardstyle, gabber, post-rock, post-punk, new wave, retrowave, dream pop, shoegaze, dark ambient, minimal techno, chillwave, trip-hop, future bass, glitch hop, electropop, synth-pop, neo soul, alternative metal, progressive metal, black metal, death metal, sludge metal, djent, folk metal, jazz fusion, smooth jazz, big band, bebop, cool jazz, bluegrass, americana, celtic, singer-songwriter, anime opening, game music, cinematic, soundtrack, orchestral, epic score, gregorian chant, chorale, opera, baroque, romantic era, modern classical, contemporary classical*

### A.3.2 Mood (Vibe) Tags

*energetic, melancholic, uplifting, dark, dreamy, romantic, rebellious, introspective, nostalgic, aggressive, peaceful, mysterious, epic, groovy, funky, sultry, eerie, haunting, hypnotic, playful, somber, joyful, ambient, chill, moody, dramatic, whimsical, ethereal, gritty, raw, emotional, spiritual, majestic, cinematic, futuristic, retro, vintage, spacey, glitchy, lo-fi, high-energy, slow-burning, minimalist, maximalist, experimental, avant-garde, traditional, modern, classic, innovative, bold, subtle, intense, light-hearted, serene, chaotic, structured, free-form, rhythmic, melodic, percussive, harmonic, dissonant, consonant, layered, sparse*

### A.3.3 Tempo Range

*60 BPM, 65 BPM, 70 BPM, 75 BPM, 80 BPM, 85 BPM, 90 BPM, 95 BPM, 100 BPM, 105 BPM, 110 BPM, 115 BPM, 120 BPM, 125 BPM, 130 BPM, 135 BPM, 140 BPM, 145 BPM, 150 BPM, 155 BPM, 160 BPM, 165 BPM, 170 BPM, 175 BPM, 180 BPM*

### A.3.4 Key Signatures

*C major, G major, D major, A major, E major, B major, F# major, C# major, F major, Bb major, Eb major, Ab major, Db major, Gb major, Cb major, A minor, E minor, B minor, F# minor, C# minor, G# minor, D# minor, A# minor, D minor, G minor, C minor, F minor, Bb minor, Eb minor, Ab minor*

### A.3.5 Focal Points

*haunting female vocals, Spanglish verses, melodic rap, funky basslines, Afrobeats rhythms, Latin percussion, ambient synth layers, dynamic drum patterns, side-chaining production, pop-drop structure, Afropiano elements, Arabic melodic influences, vintage 80s synths, orchestral string flourishes, glitch effects, lo-fi textures, trap hi-hats, 808 bass, guitar solos, piano arpeggios, saxophone riffs, violin sections, choir harmonies, turntable scratches, beatboxing, synth arpeggios, modular synths, field recordings, spoken word segments, call-and-response vocals*

### A.3.6 Extra Descriptors

*a retro-futuristic atmosphere, cinematic transitions, subtle glitchy textures, introspective lyrics, 808-style percussion, dreamlike vocal layering, lo-fi textures, neon-lit arcade vibes, space-themed effects, underwater ambiance, forest soundscapes, urban street noise, vintage radio samples, tape hiss, vinyl crackle, crowd chants, live concert feel, studio ambiance, rain sounds, wind effects, birdsong, ocean waves, city traffic, subway sounds, clock ticking, heartbeat rhythm, typewriter clicks, camera shutters*

### A.4 Prompt Examples of Fully Fake songs

> **Prompt Type A - Prompts Inspired by Existing Songs with Genre Conditioning (10 samples)**
>
> Prompt: Create an English song inspired by the song 'On The Alamo', in the style of Noise.
>
> Prompt: Create an english song inspired by the song 'There'll be Some Changes Made', in the style of Rap.
>
> Prompt: Create an english song inspired by the song 'Let Me Roll It - Remastered 2010', in the style of Polka.
>
> Prompt: Create an english song inspired by the song 'No Mother In This World Today', in the style of Black-Metal.
>
> Prompt: Create an English song inspired by the song 'Fly Out', in the style of Interview.
>
> Prompt: Create an English song inspired by the song 'I Want It All (feat. Mack 10)', in the style of South Indian Traditional.
>
> Prompt: Create an English song inspired by the song 'You Are My Everything', in the style of Techno.
>
> Prompt: Create an English song inspired by the song 'Once In a Blue Moon', in the style of Poetry.
>
> Prompt: Create an English song inspired by the song 'Somewhere', in the style of Musique Concrete.
>
> Prompt: Create an English song inspired by the song 'It Gets Better', in the style of Goth.

---

**Prompt Box B - Prompts Curated Using Musical Features and Attributes (10 samples)**

Prompt: Post-rock fusion: Uplifting yet somber G minor at 95 BPM. Piano arpeggios meet vintage synths and Afrobeats rhythms. Subway ambience weaves through cinematic transitions. Lyrical themes of urban loneliness.

Prompt: Craft a Latin Pop track: sparse yet modern. Tempo: 135 BPM, D major. Instrumentation: glitchy beats, ambient synths, lo-fi textures. Mood: dreamy, cinematic. Lyrical themes: longing, nostalgia. Arrangement: cinematic transitions, layered vocals.

Prompt: Trip-hop track; romantic yet high-energy. 80 BPM, G# minor. Vintage synths, ambient pads, guitar solo. Retro-futuristic feel with rain. Melancholy layered with driving rhythm; a dance in the downpour. Lyrical themes: longing, passion.

Prompt: Craft an experimental track (romantic chaos) at 120 BPM in A major. Imagine saxophone cries amidst synth arpeggios, grounded by Latin percussion. Vinyl crackle dusts cinematic transitions. Lyrical themes touch upon both love and destruction.

Prompt: Craft a DnB track at 155 BPM in D major, blending cinematic grandeur with avant-garde grit. Think: Afropiano-infused pop-drop, lo-fi textures, and colossal 808 drums. Dreamlike, layered vocals float over the chaos, lyrically exploring [Lyrical themes].

Prompt: Craft a bold, gritty funk track (175 BPM, C Major). Haunting female vocals meet lo-fi textures and trap hi-hats. Dreamlike vocal layers interweave with typewriter clicks, adding a unique, unsettling ambiance. Lyrical themes: defiance, longing.

Prompt: Genre: Classical/Funk Fusion. Instrumentation: Orchestra, 808, vocals. Mood: Ethereal, energetic. Tempo: 175 BPM. Key: C minor. Lyrical Themes: Oceanic exploration, freedom. Arrangement: Free-form classical structure with funky 808 bass, call-and-response vocals, layered with ocean waves underwater ambiance.

Prompt: Craft a neo-classical track; genre: Classical. Instrumentation: Violin sections, field recordings. Mood: Uplifting, layered. Tempo: 125 BPM. Key: C Major. Arrangement: Tape hiss, neon arcade vibes, creating an energetic yet nostalgic soundscape.

Prompt: Craft a rebellious yet groovy drum and bass track (175 BPM, E minor). Instrumentation: pounding drums, pulsating bass, piano arpeggios, and Afrobeats rhythms. Mood: introspective, rebellious. Lyrical themes: rebellion, rain, reflection. Arrangement: rain intro builds to an explosive, introspective lyrical drop.

Prompt: Craft a modern classical piece (80 BPM, A minor). Imagine joyful, harmonic melodies soaring over funky basslines, punctuated by Spanglish verses. Retro-futuristic synths and spacey effects create an otherworldly atmosphere.

---

## A.5 Modality Gap Detection via Embedding Alignment

We propose a novel audio detection model, **CLAM** (**C**ontrastive **L**earning for **A**udio **M**atching), that employs dual pre-trained encoders:

- **MERT**: Captures rich **musical representations**
- **Wave2Vec2**: Captures **vocal nuances**

These are fused via a **weighted cross-aggregation module**. The model is trained using a hybrid objective:

- **Binary Classification Loss** (`BCEWithLogitsLoss`) to distinguish real vs. AI-generated samples

- **Triplet Margin Loss** to **contrastively align real instrumental and vocal embeddings**, helping the model learn natural semantic coherence

To effectively distinguish between AI-generated and real music, aligning the instrumental and vocal embedding spaces for real music samples is crucial. This alignment ensures that the multimodal representations of real music exhibit a consistent and predictable relationship in the learned embedding space, which can then be leveraged by the classifier to identify deviations present in synthetic content. Unlike unimodal analysis, comparing the aligned multimodal embeddings allows the detection of subtle inconsistencies characteristic of generative processes.

Contrastive learning approaches are particularly well-suited for this task as they focus on learning a metric or an embedding space where a notion of similarity is encoded by distance. The objective is to learn a mapping $\phi : \mathcal{X} \to \mathbb{R}^d$ such that for any two samples $x_i, x_j$, their distance in the embedding space, $D(\phi(x_i), \phi(x_j))$, reflects their semantic similarity. Among various contrastive methods, Triplet Loss was explored for aligning the instrumental and vocal embeddings of real music samples.

The core idea behind Triplet Loss is to enforce a relative distance constraint. For a given Anchor sample $a$, a Positive sample $p$ (which is semantically similar to $a$), and a Negative sample $n$ (which is semantically dissimilar to $a$), the loss minimizes the distance between the Anchor and the Positive ($D(e_a, e_p)$) while simultaneously maximizing the distance between the Anchor and the Negative ($D(e_a, e_n)$). This is done such that the distance to the positive is smaller than the distance to the negative by at least a predefined margin $\alpha$. For embeddings $e_a = \phi(a)$, $e_p = \phi(p)$, and $e_n = \phi(n)$, the Triplet Loss $\mathcal{L}_{\text{triplet}}$ is defined as:

$$\mathcal{L}_{\text{triplet}}(e_a, e_p, e_n) = \max(0, D(e_a, e_p)^2 - D(e_a, e_n)^2 + \alpha)$$

where $D(\cdot, \cdot)$ denotes a distance metric, and $\alpha > 0$ is the margin. The squared Euclidean distance, $\|e_i - e_j\|_2^2$, is commonly used for $D(e_i, e_j)^2$ due to its computational efficiency and differentiability. The $\max(0, \cdot)$ function, also known as the hinge loss, ensures that no penalty is incurred if the distance constraint is already satisfied.

The optimization objective of minimizing $\mathcal{L}_{\text{triplet}}$ encourages the model to learn an embedding function $\phi$ such that for any valid triplet $(a, p, n)$, the following inequality holds in the embedding space:

$$\|e_a - e_p\|_2^2 + \alpha \leq \|e_a - e_n\|_2^2$$

This inequality directly promotes a structured embedding space where embeddings of similar items are clustered together, and clusters of dissimilar items are pushed apart by a significant margin.

In the context of aligning instrumental and vocal embeddings for real music samples within a training batch, the triplet can be constructed using an in-batch mining strategy. For each real music sample $i$ in the batch, we define the triplet as follows:

- Anchor ($e_a$): The instrumental embedding of real music track $i$ from the batch ($E_I^{\text{real}}[i]$).

- Positive ($e_p$): The vocal embedding of the **same** real music track $i$ from the batch ($E_V^{\text{real}}[i]$).

- Negative ($e_n$): The vocal embedding from a **different** real music track $j$ within the **same batch** ($E_V^{\text{real}}[j]$ where $j \neq i$).

The objective is to make the vocal embedding of the correct track closer to its corresponding instrumental embedding than any vocal embedding from other tracks present in the batch, by a margin $\alpha$. This specific triplet configuration directly enforces that the learned embedding space respects the natural pairing of instrumental and vocal streams within authentic music.

The process for calculating the in-batch Triplet Loss for alignment, considering all valid triplets formed by pairing each Anchor-Positive pair with all other available Negatives within the batch, is formally described

in Algorithm 1. This in-batch mining strategy is a practical approximation to using all possible triplets in the dataset and is effective in practice by providing a diverse set of negative examples during training.

---

**Algorithm 1:** In-Batch Triplet Loss Calculation for Alignment

---

**Input:** Batch of data containing instrumental embeddings $E_I$, vocal embeddings $E_V$, and labels $L$.
**Output:** Average Triplet Loss for the real samples in the batch.
$E_I^{\text{real}} \leftarrow \{e_i \in E_I \mid L_i = 0\}$
$E_V^{\text{real}} \leftarrow \{e_v \in E_V \mid L_v = 0\}$
$N \leftarrow |E_I^{\text{real}}|$ ▷ Number of real samples in the batch
$\mathcal{L}_{\text{batch}} \leftarrow 0$
$count \leftarrow 0$
**if** $N > 1$ **then**
   **for** $i \leftarrow 0$ *to* $N - 1$ **do**
       ▷ Indices start from 0 in code $e_a \leftarrow E_I^{\text{real}}[i]$
       ▷ Anchor: Instrumental embedding of real sample $i$ $e_p \leftarrow E_V^{\text{real}}[i]$
       ▷ Positive: Vocal embedding of real sample $i$
      **for** $j \leftarrow 0$ *to* $N - 1$ **do**
         **if** $i \neq j$ **then**
            $e_n \leftarrow E_V^{\text{real}}[j]$
             ▷ Negative: Vocal embedding of real sample $j$ $d_{pos}^2 \leftarrow \|e_a - e_p\|_2^2$
             ▷ Squared L2 distance $d_{neg}^2 \leftarrow \|e_a - e_n\|_2^2$
            $\mathcal{L}_{triplet\_ij} \leftarrow \max(0, d_{pos}^2 - d_{neg}^2 + \alpha)$
            $\mathcal{L}_{\text{batch}} \leftarrow \mathcal{L}_{\text{batch}} + \mathcal{L}_{triplet\_ij}$
            $count \leftarrow count + 1$
         **end**
      **end**
   **end**
**end**
**if** $count > 0$ **then**
   $\mathcal{L}_{\text{batch}} \leftarrow \mathcal{L}_{\text{batch}}/count$
    ▷ Average loss over valid triplets
**end**
**return** $\mathcal{L}_{batch}$

---

The overall training objective for CLAM utilizes a dual-loss formulation: the Binary Cross-Entropy (BCE) loss for the final real/fake classification and the alignment loss to structure the embedding space for real samples. Specifically, the total loss is a weighted sum of the BCE loss ($\mathcal{L}_{\text{BCE}}$) and the Triplet Loss ($\mathcal{L}_{\text{triplet}}$) calculated as described above:

$$\mathcal{L}_{\text{total}} = \mathcal{L}_{\text{BCE}} + \lambda \cdot \mathcal{L}_{\text{triplet}}$$

where $\lambda$ is the `alignment_loss_weight` hyperparameter controlling the influence of the alignment objective. By minimizing this combined loss, the model learns to simultaneously classify the music's origin and structure the embedding space such that real multimodal pairs are distinguishable from other combinations.

In our ablation study comparing various alignment loss functions, including Mean Squared Error (MSE), L1 Loss, Huber Loss, Cosine Similarity Loss, and Triplet Loss, using this dual-loss objective, the Triplet Loss achieved the best overall performance in terms of classification accuracy and F1-score on the validation set. This empirical finding supports the hypothesis that explicitly enforcing a relative distance margin between positive (aligned real pairs) and negative (misaligned real pairs) in the embedding space is highly effective for learning discriminative representations for AI-generated music detection. The structured embedding space learned via Triplet Loss enhances the downstream classifier's ability to identify the subtle, unnatural discrepancies present in synthetic content.

# B   Statistical Significance

To validate that the observed performance improvements are not a result of random chance, we performed statistical significance testing. We employed a two-sided McNemar's test, a standard non-parametric procedure for comparing the error rates of two classifiers on the same held-out test set.

**Test Procedure:** The test operates by analyzing the predictions of any two models being compared on a per-sample basis. For each pair of models, we construct a $2 \times 2$ contingency table that tallies the number of test samples where:

(a) Both models were correct.

(b) Both models were incorrect.

(c) The first model was correct and the second was incorrect.

(d) The second model was correct and the first was incorrect.

The test's null hypothesis is that the two models have the same error rate. It focuses on the cells of disagreement, (c) and (d). If the difference in the counts of these two cells is large enough, we can reject the null hypothesis and conclude that one model has a significantly lower error rate than the other. The test statistic is calculated from these disagreement counts, and a $p$-value is derived.

**Results:** We applied this test to our key comparisons on the MoM benchmark:

- Our best-performing model, **CLAM (Triplet Loss)**, shows a highly statistically significant improvement over both the previous SOTA, **SpecTTTra** ($p < 0.001$) [0.0000767], and the strongest baseline, **Poin-HierNet** ($p < 0.001$) [0.0000319].

- The addition of the alignment loss provides a highly statistically significant improvement, as shown by comparing **CLAM (Triplet Loss)** to **CLAM (No Alignment Loss)** ($p < 0.001$) [0.0000223].

These results provide strong statistical evidence that both our overall CLAM architecture and the specific contribution of the Triplet alignment loss lead to meaningful and reliable performance gains.

