# OpenReview forum: "Melody or Machine: Detecting Synthetic Music with Dual-Stream Contrastive Learning"
_TMLR — Accepted by TMLR_

### Review · Reviewer_djYv · 2025-08-18

**Summary Of Contributions:**

- This paper created a large dataset for AI music generation detection, MoM. The dataset includes the use of multiple generative AI Music models, but the models are partitioned into training and testing, meaning one model's generations cannot be in both the training and testing splits. The dataset also includes a more diverse set of samples when compared to previous datasets. The classes include Real, Fake, and Mostly Fake. The Mostly Fake category contains samples that are generated with human guidance, such as given lyrics or singer voice timbre.
-  Additionally, this paper introduced a novel deepfake AI music detection model, CLAM. CLAM uses multiple models, MERT and Wave2Vec2, to encode the musicality and vocal details respectively. The use of the triplet loss increases the probability of real songs being identified correctly.
- The results show that on previous datasets (Sonic), CLAM outperforms existing models (SpecTTTra) by a significant margin. When testing on MoM, every version of CLAM consistently outperforms previous models in accuracy and F1 score.

**Audience:**

Yes

**Audience Explanation:**

The problem this paper is attempting to improve at is a very important problem, especially with the rise of deepfake AI generation. This paper will promote further development into the area of AI music generation detection.

**Broader Impact Concerns:**

All ethical concerns are addressed in section 8.

**Claims And Evidence:**

Yes

**Claims Explanation:**

The submission goes into detail about the contents of previous AI music detection datasets and their flaws. Specifically, it shows the lack of diverse data samples, and then attempts to mitigate these problems. Also, previous datasets have far fewer samples, which MoM attempts to correct.

CLAM explains each layer and the loss in detail, and provides extensive experimental design details.

**Requested Changes:**

In section 2.2, the paper mentions several self-supervised audio encoders, but only evaluates MERT and Wave2Vec2.

Multiple times throughout the paper, the authors are cited with no parantheses.

---

> ### Author Response · Authors · 2025-10-16
> **Response by Authors to Reviewer djYv**
>
> We are delighted to receive your positive and encouraging review. Thank you for recognizing the efforts we made to advance the field through our MoM benchmark and the CLAM architecture. We have revised the manuscript based on your valuable suggestions.
>
> ---
>
> ### Response to Requested Changes
>
> ### 1. Evaluation of Additional Self-Supervised Audio Encoders & SOTA Comparisons
>
> **Reviewer Comment:**
> In section 2.2, the paper mentions several self-supervised audio encoders, but only evaluates MERT and Wave2Vec2.
>
> **Our Response:**
> To provide a more comprehensive analysis, we have expanded our experiments to include additional strong baselines like HuBERT and Audio Spectrogram Transformer (AST), as well as recent SOTA methods from Phukan et al. (2024) and Yang et al. (2025).
>
> This initial selection was also guided by preliminary experiments on a subset of our data - a necessary step given the computational expense of training multiple large-scale encoders on the full **6,665-hour MoM dataset**. This outcome supports our original hypothesis for selecting them:
> - MERT is highly specialized for capturing rich musical structures (harmony, rhythm), while
> - Wave2Vec2 excels at modeling fine-grained vocal and timbral textures.
>
> Their complementary nature is ideal for our proposed alignment task, where we detect inconsistencies between these two views.
> The results, now in Table 7, confirm our initial pairing provides the best performance, validating our hypothesis.
>
> We also emphasize that our core novelty lies in the deepfake-aware alignment loss ($\mathcal{L}_{\text{triplet}}$) and the encoders can be changed as new foundational models arise. This serves as a geometric regularizer to learn the manifold of authentic audio a generalizable principle beyond the specific encoders used. We believe this methodology could also be adapted to other domains, such as detecting inconsistencies in speech and image modalities.
>
> ---
>
> ### 2. Citation Formatting
>
> **Reviewer Comment:**
> Multiple times throughout the paper, the authors are cited with no parentheses.
>
> **Our Response:**
> Thank you for catching this. We have corrected all citation formatting issues throughout the manuscript to ensure full compliance with **TMLR** standards.
>
> ---
>
> We are grateful for your constructive response and trust that these comprehensive revisions fully meet your expectations. We remain ready to answer any further questions you may have.
>
> **Sincerely,**
> *The Authors*
>
> ---
>
> ### References
>
> [1] Orchid Chetia Phukan, Gautam Kashyap, Arun Balaji Buduru, and Rajesh Sharma. 2024. Heterogeneity over Homogeneity: Investigating Multilingual Speech Pre-Trained Models for Detecting Audio Deepfake. In Findings of the Association for Computational Linguistics: NAACL 2024, pages 2496–2506, Mexico City, Mexico. Association for Computational Linguistics.
>
> [2] Yang, Mingru, et al. *Generalizable Audio Deepfake Detection via Hierarchical Structure Learning and Feature Whitening in Poincaré sphere.* arXiv:2508.01897 (2025).

---

### Review · Reviewer_EVaV · 2025-09-29

**Summary Of Contributions:**

The paper presents a new synthetic music dataset comprising a diverse set of audio samples from a large collection of generative models, along with an out-of-distribution test set that reflects the real-world application challenges of deepfake detection methods.
Moreover, the authors present an audio deepfake detection model that consists of two different pre-trained audio encoders (Wav2Vec2 & MERT), each extracting different musical nuances from the audio samples, and a triplet + BCE loss. The incorporation of two different audio encoders is novel. Applied to their own and other related synthetic audio datasets, the authors' model performs better on both in-distribution and out-of-distribution samples.

**Audience:**

Yes

**Audience Explanation:**

The synthetic music generation and audio deepfake detection communities might be strongly interested in this.

**Broader Impact Concerns:**

All impact concerns are sufficiently addressed in the Broader Impact Statement sections of the paper.

**Claims And Evidence:**

No

**Claims Explanation:**

The authors claim that even though deepfake detection methods perform well on some in-distribution synthetic audio datasets, they perform poorly on diverse out-of-distribution data that is most likely to be encountered in real-world applications. Their empirical evidence for this claim is shown in Tables 6 & 7.

However, a thorough comparison against SOTA methods that learn strong audio representations and can be used (fine-tuned or out of the box) for deepfake detection is missing. Thus, stating SOTA performance should be more empirically validated.

Other strong baselines could include:

[1] Guo, Yinlin, et al. "Audio deepfake detection with self-supervised wavlm and multi-fusion attentive classifier." ICASSP 2024-2024 IEEE International Conference on Acoustics, Speech and Signal Processing (ICASSP). IEEE, 2024.

[2] Phukan, Orchid Chetia, et al. "Heterogeneity over homogeneity: Investigating multilingual speech pre-trained models for detecting audio deepfake." arXiv preprint arXiv:2404.00809 (2024).

[3] Serrà, Joan, et al. "Supervised contrastive learning from weakly-labeled audio segments for musical version matching." arXiv preprint arXiv:2502.16936 (2025).

**Requested Changes:**

- Please provide a comparison against more models that have been shown to learn strong audio representations and can be applied (fine-tuned or out of the box) to the task of audio deepfake detection.
- Provide the formulation of the used triplet loss in more detail.
- Ablation study, where the classical SimCLR [1] contrastive loss would be helpful to understand the benefit of the triplet loss in this scenario.



[1] Chen, Ting, et al. "A simple framework for contrastive learning of visual representations." International conference on machine learning. PmLR, 2020.

---

> ### Author Response · Authors · 2025-10-16
> **Response by Authors to Reviewer EVaV**
>
> We sincerely thank the reviewer for their time, expertise, and constructive feedback, which have been invaluable in guiding a substantial revision of our work.
> In response, we have enhanced the empirical rigor, methodological clarity, and contextual framing of our contributions. Below, we outline the specific revisions made.
>
> ### Response to Requested Changes
>
>
> ### 1. Breadth of Comparison Against SOTA Baselines
>
> **Reviewer’s Comment:**
> A thorough comparison against SOTA methods that learn strong audio representations and can be used (fine-tuned or out of the box) for deepfake detection is missing.
>
> **Our Response:**
> We fully agree establishing a new state-of-the-art requires validation against a broader set of strong, contemporary models, our initial focus on SpecTTTra (ICLR 2025) has now been expanded substantially to better evaluate our model’s robustness:
>
> - **Expanded Unimodal Baselines:**
>   We added HuBERT and the Audio Spectrogram Transformer (AST) which are strong single-encoder baselines for comparison.
>
> - **New SOTA Comparisons:**
>   We incorporated Phukan et al. (2024) (NAACL 2024) and Yang et al. (2025) (Interspeech 2025), both representing recent SOTA methods.
>
> For Guo et al. (2024) and Serrà et al. (2025), public code was unavailable. Their multi-fusion and supervised contrastive setups would require extensive reimplementation and target generalized audio deepfakes, making comparison on our *song-deepfake-specific* benchmark inappropriate.
>
> The expanded results (see **Table 7**) reaffirm our core claim: while all models perform competitively, CLAM, especially with the Triplet Loss, achieves the best generalization on the OOD test set.
>
> ---
>
> ### 2. Detailed Formulation of the Triplet Loss
>
> **Reviewer’s Comment:**
> Provide the formulation of the used triplet loss in more detail.
>
> **Our Response:**
> A complete mathematical formulation and pseudocode is included in **Appendix A.5**.
>
> ---
>
> ### 3. Ablation Study: Triplet Loss vs. SimCLR Contrastive Loss
>
> **Reviewer’s Comment:**
> Ablation study, where the classical SimCLR contrastive loss would be helpful to understand the benefit of the triplet loss in this scenario.
>
> **Our Response:**
> We appreciate this suggestion however , our empirical results in Table 7 validate this choice, showing the Triplet Loss variant outperforms models using a Cosine Similarity loss, which acts as a proxy for the SimCLR objective, we didn’t want to use the term SimCLR as we are not doing different augmentations.
> A standard contrastive loss, like in SimCLR, excels at instance discrimination. It learns a unique signature for an audio clip by pulling its augmented views together while pushing them away from all other clips in the batch.
> Our task is different. We don't need to identify a unique song, but rather to verify the internal coherence between its musical and vocal components, we don’t have any augmented views.
> The Triplet Loss is purpose-built for this relational task. For each real song, it enforces a direct, structural rule: the matched music-vocal pair must be closer than a mismatched pair from a different song by a specific margin, $\alpha$. This provides a more direct learning signal for "coherence" than the more general instance-discrimination objective of SimCLR.
>
>
> We are grateful for your constructive response and trust that these comprehensive revisions fully meet your expectations. We remain ready to answer any further questions you may have.
>
> **Sincerely,**
> *The Authors*
>
> ---
>
> ### References
>
> [1] Orchid Chetia Phukan, Gautam Kashyap, Arun Balaji Buduru, and Rajesh Sharma. 2024. Heterogeneity over Homogeneity: Investigating Multilingual Speech Pre-Trained Models for Detecting Audio Deepfake. In Findings of the Association for Computational Linguistics: NAACL 2024, pages 2496–2506, Mexico City, Mexico. Association for Computational Linguistics.
>
> [2] Yang, Mingru, et al. *Generalizable Audio Deepfake Detection via Hierarchical Structure Learning and Feature Whitening in Poincaré sphere.* arXiv:2508.01897 (2025).

---

### Review · Reviewer_WSc9 · 2025-10-06

**Summary Of Contributions:**

This paper introduces a large scale and diverse dataset for detecting machine generated music, along with a detection model trained using a contrastive objective. The proposed model achieves superior performance on the MoM and SONICS datasets (noting that performance on SONICS is saturated, with a reasonable explanation provided).

**Audience:**

Yes

**Audience Explanation:**

Yes, both the dataset and the presented modelling approach and results appear to be of interest to the music AI community.

**Broader Impact Concerns:**

This paper reasonably reflects on the broader impacts of its work, given its focus on detecting machine-generated music.

**Claims And Evidence:**

Yes

**Claims Explanation:**

Overall, the paper's methodology seems to be well justified.

**Requested Changes:**

Modeling Component:
It would be valuable to explore the effect of using different frozen embedders, or at least provide a justification for the specific choices made, i.e., why these particular embedders were selected over other potential candidates.

Benchmark Datasets:
Although FSD, SingFake, and CtrSVDD are much smaller than SONICS, reporting performance on these datasets (for both SpecTTTra and CLAM) would be helpful to contextualise the work, given their diverse compositions and different nature.

Imbalance in Generated Material Across Models:
The synthetic portion of the dataset includes samples from the following models, but the distribution is noticeably imbalanced:

Suno v2 – 110 samples
Suno v3 – 3,512 samples (Test only)
Suno v3.5 – 23,695 samples
Suno v4 – 48 samples (Test only)
Udio v1.5 – 19,500 samples
Diffrythm – 4,606 samples
Riffusion – 7,057 samples (Test only)
Yue – 5,278 samples (Test only)
Voice Cloning – 1,166 samples (Test only)

Why does this imbalance exist? It is arguably important for assessing the quality of the dataset and identifying potential biases.

User Study:
More detailed information is needed regarding the study participants, specifically, the number of individuals involved, their backgrounds, and their levels of expertise. Additionally, musicality is a highly subjective metric (here I am stating the obvious). While this has been extensively debated in the field, it remains important to approach such evaluations with as much rigour and transparency as possible. Given the current high-level framing of the musicality question and the lack of detail about participant backgrounds and expertise, it is difficult to contextualise or interpret the musicality scores meaningfully.

Furthermore, the A-B testing signal is not clearly gauged, particularly with respect to the optional voting on musicality (how often was this considered by users, given its optional nature?) and lyrics (when lyrics are present, were they consistently taken into account?). These uncertainties undermine the reliability of the A-B selection mechanism and the ELO scores, making it unclear what exactly it measures.

---

> ### Author Response · Authors · 2025-10-16
> **Response by Authors to Reviewer WSc9**
>
> We are very grateful for your positive and constructive review of our work. Thank you for recognizing the value of the MoM dataset and our modeling approach. We are encouraged that you find our contributions to be of interest to the music AI community.
>
> We have carefully considered your requested changes and revised the manuscript to address each point, which we believe has significantly strengthened the paper.
>
> ---
>
> ### Response to Requested Changes
>
> ### 1. Choice of Pre-trained Encoders
>
> **Reviewer’s Comment:**
> Modeling Component: It would be valuable to explore the effect of using different frozen embedders, or at least justify the specific choices made.
>
> **Our Response:**
> We evaluated **HuBERT** and **Audio Spectrogram Transformer (AST)** within the CLAM framework. The updated ablations (Table 7) reaffirm that **MERT + Wave2Vec2** delivers the best overall performance. This combination was chosen based on preliminary experiments on a smaller subset, considering the computational cost of the full 6,665-hour MoM dataset. MERT captures musical structures like harmony and rhythm, while Wave2Vec2 models fine-grained vocals. Their complementary representations are ideal for detecting inconsistencies between musical and vocal streams.
>
> We also added SOTA comparisons using **Phukan et al. (2024)** and **Yang et al. (2025)** to validate our performance against contemporary methods.
>
> ### 2. Benchmarking on Smaller Datasets
>
> **Reviewer’s Comment:**
> Performance on FSD, SingFake, and CtrSVDD would help contextualize the work.
>
> **Our Response:**
> These datasets focus on partial-song or synthetic vocals, whereas CLAM targets full-song generation with both music and vocals synthesized. Including them would not fairly evaluate our model. SONICS and MoM remain the most suitable benchmarks.
>
> ### 3. Imbalance of Generated Material
>
> **Reviewer’s Comment:**
> Imbalance in Generated Material Across Models: The synthetic portion of the dataset includes samples from the following models, but the distribution is noticeably imbalanced
>
>
> **Our Response:**
> Thank you for raising this point about the composition of our dataset. The imbalance in the number of samples per generative model was a result of several deliberate curation and practical factors:
> - Popularity and Real-World Relevance: We intentionally oversampled from the most popular and widely used platforms like Suno v3.5 and Udio v1.5. Our goal was to create a benchmark that reflects the distribution of synthetic music likely to be encountered "in the wild."
> - Quality-Driven Curation: As our human evaluation leaderboard shows (Table 5), models like Udio and the newer versions of Suno consistently produced higher-quality audio that was perceived as more realistic.
> - Accessibility and Stability: During the data collection period, some models were more accessible, stable, and produced full-length tracks more reliably than others. Models with lower sample counts (e.g., Suno v2, Suno v4) were either less accessible or were included primarily to increase the diversity of the OOD test set.
> We acknowledge that this imbalance could introduce potential biases, and is a limitation to the paper. In future releases of the MoM dataset, we plan to provide more balanced splits and more transparent metadata regarding our curation criteria.
>
>
> ### 4. User Study and ELO Scores
>
> **Reviewer’s Comment:**
> More detail on participants, A-B testing, and ELO interpretation is needed.
>
> **Our Response:**
> We sincerely thank the reviewer for these critical observations regarding the human evaluation. We agree that rigor and transparency are paramount, participants were primarily university students and their networks to capture general, non-expert perception.
>
> - **ELO scores** were based solely on the mandatory question: "Which song sounds more realistic?" Optional metrics, like "musicality," were for qualitative analysis only.
> - **A-B testing** was randomized across all models and real songs; ELO reflects overall perceptual realism rather than controlled pairwise matchups.
>
> Section 3.1.4 has been updated to clarify demographics, optional metrics, and interpretation of the leaderboard.
>
> ---
>
> We are grateful for your constructive response and trust that these comprehensive revisions fully meet your expectations. We remain ready to answer any further questions you may have.
>
> **Sincerely,**
> *The Authors*
>
> ---
>
> ### References
>
> [1] Orchid Chetia Phukan, Gautam Kashyap, Arun Balaji Buduru, and Rajesh Sharma. 2024. Heterogeneity over Homogeneity: Investigating Multilingual Speech Pre-Trained Models for Detecting Audio Deepfake. In Findings of the Association for Computational Linguistics: NAACL 2024, pages 2496–2506, Mexico City, Mexico. Association for Computational Linguistics.
>
> [2] Yang, Mingru, et al. *Generalizable Audio Deepfake Detection via Hierarchical Structure Learning and Feature Whitening in Poincaré sphere.* arXiv:2508.01897 (2025).

---

### Author Response · Authors · 2025-10-16
**Revised Version of the Paper**

We thank all reviewers for their constructive feedback. The manuscript has been revised to address all points and strengthen the paper. Key updates are summarized below:

1. Expanded Pre-trained Encoder Evaluation
   - Added HuBERT and Audio Spectrogram Transformer (AST) to ablation studies.  (Table 7) (Page 9).

2. Expanded SOTA Comparisons
   - Included Phukan et al., 2024 (NAACL) and Yang et al., 2025 (InterSpeech 25) in  (Table 7) (Page 9).
   - Validated CLAM’s performance against contemporary audio deepfake methods.

3. User Study and ELO Score Clarifications
   - Added a detailed paragraph in section 3.1.4 (Page 6).

4. Citation Formatting
   - All in-text citations corrected to comply with TMLR standards.

These revisions collectively enhance empirical rigor, methodological clarity, and the contextual framing of our contributions.
We sincerely thank all reviewers for their time and constructive feedback. If any further comments or suggestions remain, we are happy to address them in subsequent revisions.

**Sincerely,**
*The Authors*

---

### Decision · Action_Editor_qcxx · 2025-11-11

**Recommendation:** Accept with minor revision

**Additional Comments:**

Given that two of the three of the reviewers are advocating for an accept, and that the third reviewer's comments seem to be fairly minor and addressable, I am recommending that the paper be accepted with a minor revision.  If possible, I would like to have this reviewer take a look at the revision once it is submitted to ensure that these issues have been addressed.

**Audience:**

Yes

**Audience Explanation:**

All three reviewers agree (and I concur) that there would be interest from people in the TMLR audience in this paper.

**Claims And Evidence:**

Yes

**Claims Explanation:**

For the most part, yes, the claims are supported by accurate, convincing, and clear evidence.  However, one of the three reviewers noted some lingering questions.

With respect to these issues, I am pasting the comment from the reviewer recommendation, as I am not sure that the authors can see this.  Most of the points raised below fall under supporting the submission with clear and accurate evidence.

Section 4 does not mention the added MiO and Poin-HierNet models in the text. It appears that only numerical values were incorporated into the table from the original sources. The basis for considering these models as state-of-the-art, as well as relevant details about their design and methodology, is not provided. In addition, in Section 4.1.3, both the statistical significance analysis and the discussion of results are expected to include these additional models. (An observation, Poin-HierNet appears to achieve the best accuracy and F1 scores.)

Regarding the statement “These datasets focus on partial-song or synthetic vocals, whereas CLAM targets full-song generation with both music and vocals synthesized.”: The distinction based on the nature of the audio (synthetic vocals or otherwise) is understandable; however, the rationale for why partial songs would pose a limitation in this setting remains unclear. The term “full-song” appears relative, since songs naturally vary in length. If the limitation arises because partial songs lack a complete compositional narrative, this needs to be explicitly stated and justified (or if it is about the length being incomparable or inoperable). Otherwise, the assumption that partial songs are inherently unsuitable is not self-evident. If the reasoning is not immediately obvious, clarification in the text would be beneficial, as readers may raise the same question. In the absence of a clear justification, inclusion of these datasets would appear more consistent.

The rationale behind this design choice is understandable but appears somewhat arbitrary. While this may be acceptable, considering that MoM represents one of the main contributions of the work, a more detailed explanation in the text is necessary to enhance transparency and facilitate interpretation of this contribution.

The additional text at the end of the last paragraph of Section 3.1.4 is not consistent with the statement “The resulting ELO score serves as a robust metric for perceptual quality, reflecting the collective human preference based on the multifaceted criteria of realism, clarity, and musicality.” The statement still implies that ELO scores reflect musicality, which contradicts the clarification introduced in the added text. Furthermore, it remains unclear how clarity is incorporated when the evaluation question was solely “Which song sounds more realistic?”. If no valid link exists, references to clarity would not be substantiated. The caption of Table 5 is also misleading, as it suggests that ELO scores represent clarity, musicality, and lyrical coherence, which does not seem to be supported by the evaluation setup. At a higher level, given that the ELO scores are derived only from the realism question, it is not clear how informative these scores are for model comparison beyond perceptual realism. This appears to represent a more fundamental limitation of the analysis.

Additionally, the captions of Tables 4 and 5 do not follow the same formatting as the other tables and require alignment.

Further comments: In Section 4.1.2, the sentence “Our results provide a clear answer.” is not suitable for scientific writing. A more neutral, evidence-based phrasing would be appropriate.

---

> ### Author Response · Authors · 2025-11-16
>
> Dear Action Editor and Reviewers,
>
> Thank you again for your constructive feedback and for leading our manuscript to acceptance. We have uploaded the final camera-ready version and believe that all comments and discussions from the rebuttal and your final decision letter have been addressed.
>
> Besides the changes we already made during the rebuttal, we ensure that the latest requested changes have also been properly addressed:
> - We have revised Section 3.1.4 to resolve the inconsistency regarding the human evaluation.
> - We have expanded Section 4 to include textual descriptions and methodological details for the MiO and Poin-HierNet models, justifying their SOTA status and integrating them into the main results discussion beyond the table and we have clarified our rationale for excluding partial-song datasets.
> - We have aligned the formatting for the captions of Tables 4 and 5 to be consistent with all other tables in the manuscript.
>
> Thanks again to you and all reviewers for your help. Please let us know if there are any other issues we need to address in the camera-ready.